# Enhanced Magnetocaloric Properties of Annealed Melt-Extracted Mn$_{1.3}$Fe$_{0.6}$P$_{0.5}$Si$_{0.5}$ Microwires

**Lin Luo** [1,2], **Jia Yan Law** [2,*], **Hongxian Shen** [1], **Luis M. Moreno-Ramírez** [2], **Victorino Franco** [2], **Shu Guo** [1], **Nguyen Thi My Duc** [3,4], **Jianfei Sun** [1,*] and **Manh-Huong Phan** [3,*]

1   School of Materials Science and Engineering, Harbin Institute of Technology, Harbin 150001, China
2   Department of Condensed Matter Physics, University of Seville, 41080 Seville, Spain
3   Department of Physics, University of South Florida, Tampa, FL 33620, USA
4   Department of Material Science, Faculty of Physics, The University of Danang—University of Science and Education, 459 Ton Duc Thang, Lien Chieu, Danang 550000, Vietnam
*   Correspondence: jylaw@us.es (J.Y.L.); jfsun@hit.edu.cn (J.S.); phanm@usf.edu (M.-H.P.)

**Abstract:** The highly regarded Fe$_2$P-based magnetocaloric materials are usually fabricated by ball milling, and require an additional extended annealing treatment at high temperatures (at temperatures up to 1423 K for several hours to days). In this work, we show that fabricating Mn$_{1.3}$Fe$_{0.6}$P$_{0.5}$Si$_{0.5}$ into the form of microwires attained 82.1 wt.% of the desired Fe$_2$P phase in the as-cast state. The microwires show a variable solidification structure along the radial direction; close to the copper wheel contact, Fe$_2$P phase is in fine grains, followed by dendritic Fe$_2$P grains and finally secondary (Mn,Fe)$_5$Si$_3$ phase in addition to the dendritic Fe$_2$P grains. The as-cast microwires undergo a ferro- to para-magnetic transition with a Curie temperature of 138 K, showing a maximum isothermal magnetic entropy change of 4.6 J kg$^{-1}$ K$^{-1}$ for a magnetic field change of 5 T. With further annealing, a two-fold increase in the maximum isothermal magnetic entropy change is found in the annealed microwires, which reveal 88.1 wt.% of Fe$_2$P phase.

**Keywords:** melt-extraction technique; (Mn,Fe)$_2$(P,Si) microwires; microstructure; magnetocaloric effect

## 1. Introduction

Solid-state magnetic refrigeration is considered a potential alternative to the conventional vapor-compression techniques due to its high energy efficiency and environmental friendliness [1–6]. The development of magnetic refrigerators depends on the performance of magnetic refrigerants, those having significant magnetocaloric effect (MCE) [7–9]. (Mn,Fe)$_2$(P,Si) materials are one of the most promising magnetic refrigerants due to their high magnetocaloric performance, relatively low hysteresis and abundance of the raw materials [10–12].

Regarding the synthesis procedures, it is noticeable that most of the reported (Mn,Fe)$_2$(P,Si) systems are prepared by ball milling and/or arc melting methods [13,14]. However, these methods usually require time-consuming heat treatment (from several hours to several days) for achieving homogeneous compositions and forming high fractions of the Fe$_2$P phase [9]. Alternatively, rapid solidification techniques, such as melt spinning [15,16] and droplet melting [17,18], enable the fabrication of materials with excellent compositional homogeneity and reduced impurities after short-time heat treatment. In this way, melt-extraction as a rapid solidification technique has been proposed as a promising method to develop wire-shaped magnetocaloric materials as reported for melt-extracted Gd-based microwires [19,20] and more recently high-entropy alloys [21,22], as well as medium-entropy alloys [23]. Moreover, theoretical calculations suggest that wire-shaped magnetic refrigerants with high packing factors could result in a larger working frequency compared to particles and layers [24,25]. Their micro-size diameters could improve the heat transfer of the cooling system [24]. Hence, it will be of interest to study

(Mn,Fe)$_2$(P,Si) melt-extraction microwires and their structural and magnetocaloric behavior. The as-cast microwires show a variable solidification structure for the Fe$_2$P phase going from fine to dendritic grains along the radial direction. An elevated amount (82.1 wt.%) of the desired Fe$_2$P phase is obtained in the as-cast Mn$_{1.3}$Fe$_{0.6}$P$_{0.5}$Si$_{0.5}$ microwires, which gives an isothermal magnetic entropy change peak value of 4.6 J kg$^{-1}$ K$^{-1}$ (for 5 T) at 145 K. A two-fold increase to 10.5 J kg$^{-1}$ K$^{-1}$ is achieved with further annealing at 1373 K for 15 min.

## 2. Materials and Methods

The MCE of (Mn,Fe)$_2$(P,Si) can be tuned by varying the Fe and Si content [26–28]. We have selected x = 0.6 in Mn$_{2-x}$Fe$_x$Si$_{0.5}$P$_{0.5}$ due to the minimized hysteresis reported in ref. [27]. In addition, for this family of compounds, slight deficiency in transition metal elements can reduce the impurities and thermal hysteresis [29,30]. Therefore, the composition of Mn$_{1.3}$Fe$_{0.6}$P$_{0.5}$Si$_{0.5}$ was chosen for researching. An ingot was prepared by arc-melting in argon atmosphere from a mixture of raw materials: Mn (99.5%), P (99.5%), FeP chunks (98%) and Si (99.999%). For accounting for the loss of Mn during arc-melting, an excess of 5 wt.% of Mn was added to the mixture. The ingot was re-melted at least six times with electromagnetic stirring then suction casted into an 8 mm diameter rod for subsequent melt-extraction into microwires. Using a melt feeding speed of 20 µm/s and a 30 cm diameter Cu wheel with a linear speed of 30 m/s, up to 7 cm long as-cast microwires with ~35 µm diameters were obtained. For annealing studies, the microwires were sealed in a quartz tube under Ar atmosphere and annealed at 1373 K for 15 min and then water quenched.

The measurements were performed, employing a large quantity from the as-cast and annealed microwires, giving that the obtained results are the average from the studied samples. Powder X-ray diffraction (XRD) experiments at room temperature using Cu-K$_\alpha$ radiation were performed to characterize for the crystal structure and phase composition of the microwires. Microstructural and compositional details were obtained on a field emission scanning electron microscope (SIGMA-500-Zeiss) with an energy dispersive spectrometer (EDS). Magnetic measurements were carried out on a Quantum Design Physical Property Measurement System (PPMS-16 T) using a standard vibrating sample magnetometer (VSM) option. A set of the microwire arrays were close packed in a sample holder with a length of ~3 mm and an inner diameter of ~1 mm. To avoid virgin effects, the samples were precooled down to 10 K. Isothermal magnetization measurements were performed using a discontinuous protocol for erasing the magnetic hysteresis prior to any measurement [31]. The isothermal magnetic entropy change ($\Delta S_{iso}$) is indirectly determined from magnetization measurements by applying the Maxwell relation:

$$\Delta S_{iso} = \mu_0 \int_0^H \left(\frac{\partial M}{\partial T}\right)_{H'} dH'. \tag{1}$$

The relative cooling power (*RCP*) are calculated using the following Equation:

$$RCP = \Delta S_{iso}^{pk} \delta_{FWHM}, \tag{2}$$

where ($\delta_{FWHM}$) is the full width at half maximum of the peak of isothermal magnetic entropy change ($\Delta S_{iso}^{pk}$).

## 3. Results and Discussion

The XRD patterns of the as-cast microwires and the Rietveld refinement results show a main phase (82.1(7) wt.%) of the desired Fe$_2$P hexagonal structure (space group of $P\bar{6}2m$) in addition to minority (Mn,Fe)$_5$Si$_3$ phase (space group of $P6_3/mcm$) as displayed in Figure 1a. As this Fe$_2$P phase contributes to the giant MCE of the Mn-Fe-P-Si family, it is worth highlighting for its phase amount obtained in the as-cast microwires since its

enhancement usually requires annealing of the as-cast ingots as reported in ref. [12]. The lattice parameters found for the $Fe_2P$ phase are $a = b = 6.09513(13)$ Å, $c = 3.45230(10)$ Å.

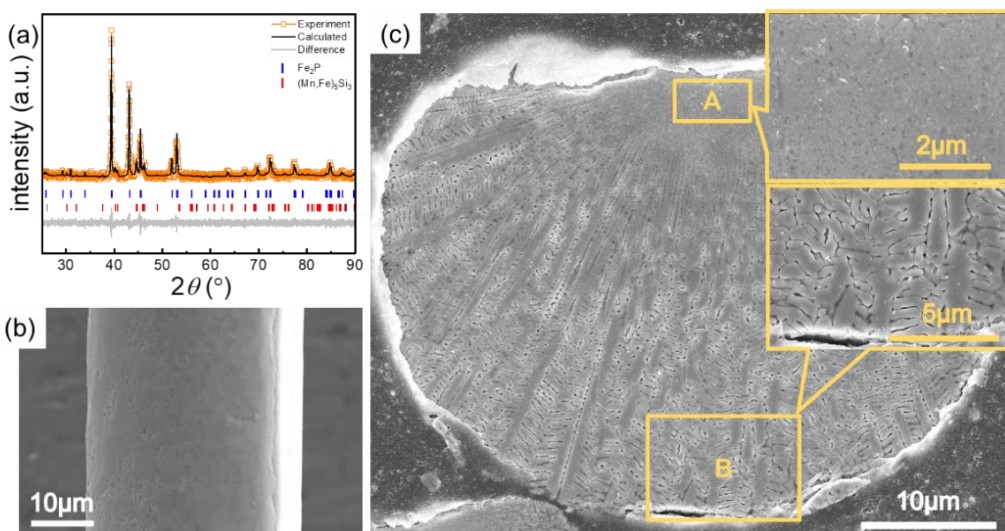

**Figure 1.** (**a**) XRD and Rietveld refinement results of the as-cast microwires. SEM of the (**b**) surface and (**c**) radial cross-section of the as-cast microwires.

The as-cast microwires are observed with a continuous and smooth surface as evidenced from their scanning electron micrograph (SEM) presented in Figure 1b. Further etching them with 4 wt.% hydrofluoric acid for 3 min, the radial cross-section of the microwires in Figure 1c reveals groove defects at the Cu-wheel side. Further magnification shows fine equiaxed grains near the wheel contact (see area A), while fine dendrites are observed (area B) further away from the wheel contact zone.

The EDS maps of the axial cross-section of the microwires presented in Figure 2a reveal an overall uniform element distribution. Only very minor fluctuations for P and Si concentrations are noticed along the radial directions when performing EDS line scans across the width and cross-section of the microwires (Figure 2b,c). These local inhomogeneities could be attributed to the existence of impurity $(Mn,Fe)_5Si_3$ phase previously observed from XRD data.

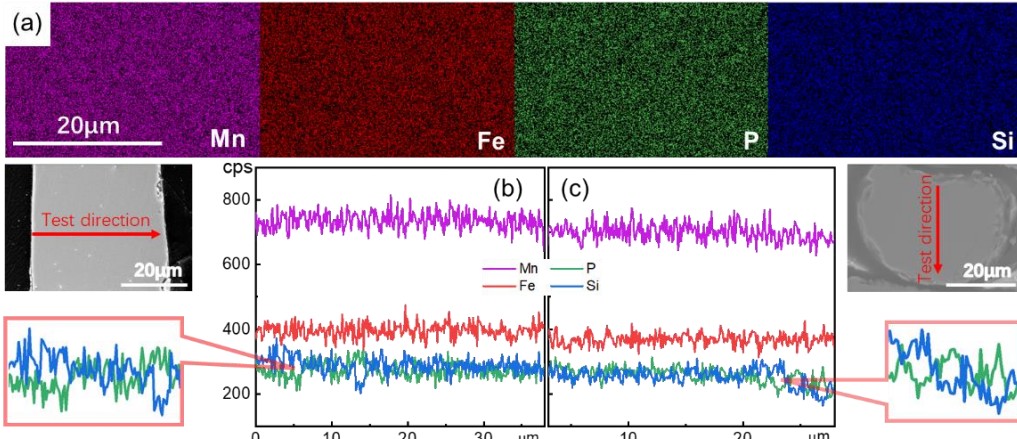

**Figure 2.** (**a**) EDS maps of axial cross-section of the as-cast microwire. EDS line scans (**b**) along the width of the microwire and (**c**) the microwire cross-section.

Figure 3a presents the $M(T)$ data of the as-cast microwires for 0.05 T and their derivatives in the inset, showing the Curie temperature ($T_C$) of 138 K. The magnetization and

specific heat measurements [32,33] are common methods for calculating the MCE performance of the materials. Here, isothermal magnetization curves were measured for determining the isothermal magnetic entropy change. Figure 3b presents the isothermal magnetic entropy change as a function of temperature for magnetic field change up to 5 T. Their peak values ($\Delta S_{\text{iso}}^{\text{pk}}$) for a field change of 2 and 5 T are found to be 1.9 and 4.6 J kg$^{-1}$ K$^{-1}$, respectively. The temperature corresponding to $\Delta S_{\text{iso}}^{\text{pk}}$ values, denoted as $T_{\text{pk}}$, is around 145 K. Furthermore, it has to be noted that despite the elevated content of the Fe$_2$P phase, the existence of the (Mn,Fe)$_5$Si$_3$ impurity phase could have a detrimental effect on the magnetocaloric properties.

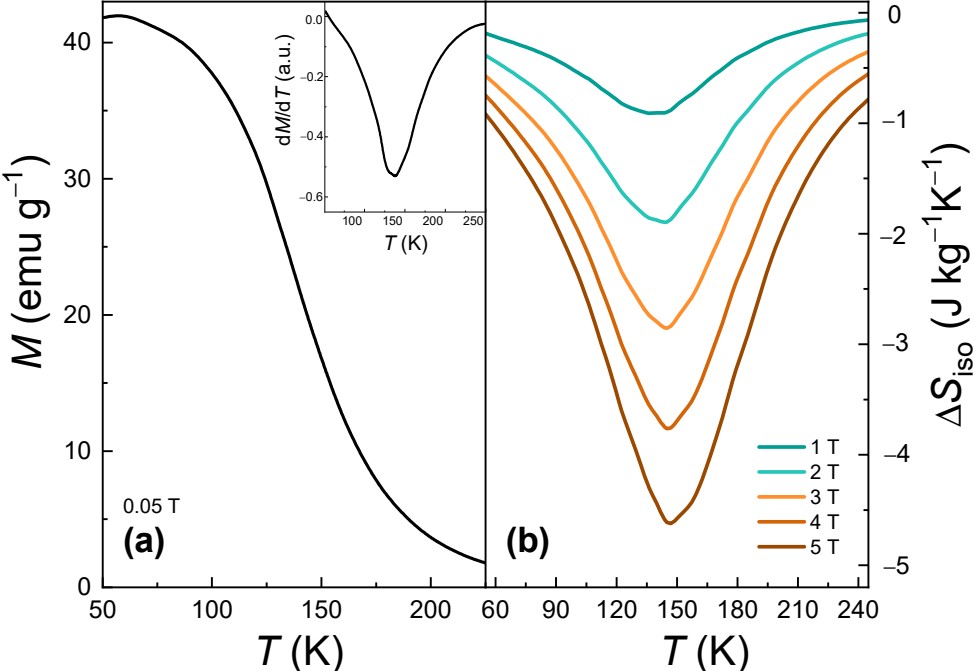

**Figure 3.** (**a**) Temperature dependence of magnetization at 0.05 T and its derivatives in the inset. (**b**) Isothermal magnetic entropy change as a function of temperature for the as-cast microwires for different magnetic field changes.

Figure 4a presents the XRD data and Rietveld refinement results of the studied microwires upon annealing, which shows that Fe$_2$P phase increases from 82.1(7) to 88.1(8) wt.%. Furthermore, the lattice parameters $a$ shrink, and lattice parameters $c$ expand after annealing, and the values are $a = b = 6.08589(10)$ Å, $c = 3.45741(10)$ Å. Figure 4b,c show the field dependence of $T_{\text{pk}}$ and $\Delta S_{\text{iso}}^{\text{pk}}$ of the as-cast and annealed microwires. It is observed that they are enhanced upon annealing and a two-fold increase from 1.9 and 4.6 to 5.1 and 10.5 J kg$^{-1}$ K$^{-1}$ (for a field change of 2 and 5 T) is achieved, which is comparable to Gd (9.8 J kg$^{-1}$ K$^{-1}$ for 5 T) [27]. These larger values of $\Delta S_{\text{iso}}^{\text{pk}}$ can be attributed to the larger amount of the Fe$_2$P phase and to the subsequent variation in the composition (closer to the nominal one after annealing). There is slightly variation in the *RCP* after annealing, as shown in Figure 4d. The *RCP* values of microwires before and after annealing are 160 are 178 J kg$^{-1}$, larger than 90 J kg$^{-1}$ of Mn$_{1.3}$Fe$_{0.65}$P$_{0.5}$Si$_{0.5}$ for a field change of 2 T [34], indicating a better cooling performance. The larger *RCP* values can be attributed to the broad ferromagnetic-paramagnetic phase transition.

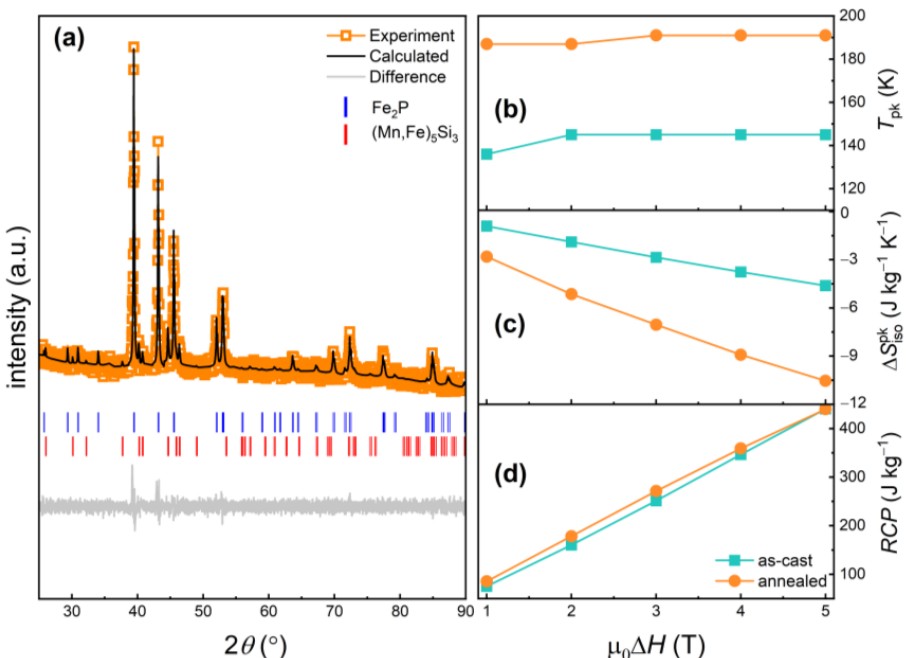

**Figure 4.** (**a**) XRD and its Rietveld refinement results; magnetic field dependence of (**b**) $T_{\mathrm{pk}}$, (**c**) $\Delta S_{\mathrm{iso}}^{\mathrm{pk}}$, and (**d**) *RCP* of the annealed microwires.

## 4. Conclusions

The $Mn_{1.3}Fe_{0.6}P_{0.5}Si_{0.5}$ microwires have been successfully fabricated by the melt-extraction technique. An elevated $Fe_2P$ phase content of 82.1 wt.% has been attained for the as-cast microwires, which is significantly improved in comparison to conventional methods. A variable solidification structure along the radial direction of the as-cast microwires is found, from fine $Fe_2P$ grains near the copper wheel contact $\rightarrow$ dendritic $Fe_2P$ grains + secondary $(Mn,Fe)_5Si_3$ phase as we advance further away from the wheel contact. The $Fe_2P$ phase content increases up to 88.1 wt.% upon further annealing, leading to an observed two-fold increase in the isothermal magnetic entropy change peak values from 1.9 and 4.6 J kg$^{-1}$ K$^{-1}$ to 5.1 and 10.5 J kg$^{-1}$ K$^{-1}$ for a field change of 2 T and 5 T. The *RCP* values of annealed melt-extracted microwires are 178 and 440 J kg$^{-1}$ (for 2 T and 5 T). These results indicated that the annealed melt-extracted microwires show improved MCE performance.

**Author Contributions:** Conceptualization, L.L. and H.S.; methodology, L.L., J.Y.L., H.S., L.M.M.-R., S.G. and N.T.M.D.; validation, L.L., J.Y.L., H.S. and L.M.M.-R.; formal analysis, L.L. and H.S.; investigation, L.L., J.Y.L., L.M.M.-R. and H.S.; resources, J.Y.L., H.S., V.F. and J.S.; data curation, L.L.; writing—original draft preparation, L.L. and N.T.M.D.; writing—review and editing, L.L., J.Y.L., L.M.M.-R., H.S. and M.-H.P.; visualization, L.L. and J.Y.L.; supervision, V.F., J.Y.L., J.S. and M.-H.P.; project administration, V.F. and J.S.; funding acquisition, V.F. and J.S. All authors read and agree to the final version of the manuscript.

**Funding:** This research was funded by the National Natural Science Foundation of China (NSFC, Nos. 51871124), grant PID2019-105720RB-I00 funded by MCIN/AEI /10.13039/501100011033 and Consejería de Economía, Conocimiento, Empresas y Universidad de la Junta de Andalucía (grant P18-RT-746). L.L. acknowledges the fellowship from China Scholarship Council (CSC, 202006120203) for Visiting Ph.D. Student program. L.M.M.-R. acknowledges a postdoctoral fellowship from Junta de Andalucía and European Social Fund (ESF).

**Institutional Review Board Statement:** Not applicable.

**Informed Consent Statement:** Not applicable.

**Data Availability Statement:** The data presented in this study are available on request from the corresponding authors.

**Conflicts of Interest:** The authors declare no conflict of interest.

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
