# Peer review of "Enhanced Magnetocaloric Properties of Annealed Melt-Extracted Mn1.3Fe0.6P0.5Si0.5 Microwires"

_metals, doi:10.3390/met12091536_

Round 1

Reviewer 1 Report

The authors have synthesized Mn1.3Fe0.6P0.5Si0.5 microwires using melt-extraction technique. Two sets of samples one without any annealing and other annealed at 1373 K for 15 minutes are prepared.  Annealing enhances Fe2P phase that in turn leads to two-fold increase in magnetocaloric effect. Crystal structure analysis is done via Rietveld refinement and EDS mapping/line scan. The manuscript is precise and well written by correlating all the results.

Only glitch is that the cooling efficiency cannot be calculated only from the magnitude of magnetic entropy change. So the authors are requested to present the relative cooling power, refrigerant capacity of the sample and compare it with relevant materials.

I recommend this paper for publication after making the above-mentioned correction.

Reviewer 2 Report

The reviewer considered the manuscript interesting and believes that can be published in this journal.

However there are some points that need to be improved:

- How many analysis of as-cast microwires were performed. The authors need to analyse at least 3 different samples in order to evaluate the repeatability.

- How was controlled the temperature dependence of magnetization at 0.05 analysis? And, how many repetitions were performed? Only one is not enough. The same comment for XRD analysis.

Reviewer 3 Report

The authors took up the topic of work: "Enhanced magnetocaloric properties of annealed melt-extracted Mn1.3Fe0.6P0.5Si0.5 microwires". Unfortunately, I do not see any special innovation in the method, apart from the faster time to receive the material.

- The authors did not make a careful literature review, as evidenced by the small number of works cited in the Introduction.

- The authors give isothermal magnetic entropy as a function of rate of temperature, but nowhere do they give specific heat as a function of temperature. Please see what it's like at work: https://doi.org/10.1016/j.actamat.2021.117437

- The XRD analysis is modest, while Figures 1a and 4a are identical.

- About structural studies depending on Fe content: https://doi.org/10.1016/j.jeurceramsoc.2021.06.018. Have the authors studied the relationships depending on the parameters of Mn and Fe and such stoichiometry turned out to be the most appropriate for the authors' research, or is it a randomly selected stoichiometry?

- Perhaps it is also worth thinking about TEM research, e.g. similarly to nanoduts at work: https://doi.org/10.1021/jp048163n.

- Conclusions need to be refined. The change in isothermal magnetic energy for the 5 T field is given. And as for other magnetic fields - have the authors undertaken such research? (not just this one field). There is no word in the MCE conclusions (what is in the title and what is the main topic of the manuscript).

There are also a lot of minor, insignificant errors in the manuscript:

- "wt.%" should be wt% (no dot)

- line 84 "in literature" should be in ref.

- lines 58-59 - there is no point in using Si raw material with a purity of 99.999%, if the remaining raw materials have much lower purities.

The manuscript requires elaboration, I cannot recommend the work in its current form.

Reviewer 4 Report

The paper can be accepted as is

Author Response

We thank the reviewer’s assessment and approval.

Round 2

Reviewer 3 Report

I have no more comments for the manuscript: "Enhanced magnetocaloric properties of annealed melt-extracted Mn1.3Fe0.6P0.5Si0.5 microwires". The authors did a literature review and answered my doubts. I recommend the paper for publication in Metals MDPI.